# MegaPose: 6D Pose Estimation of Novel Objects via Render & Compare

**Yann Labbé** [1†]    **Lucas Manuelli** [2]    **Arsalan Mousavian** [2]    **Stephen Tyree** [2]
**Stan Birchfield** [2]    **Jonathan Tremblay** [2]    **Justin Carpentier** [1]    **Mathieu Aubry** [3]
**Dieter Fox** [2,4]    **Josef Sivic** [5]

[1] ENS/Inria    [2] NVIDIA    [3] LIGM/ENPC    [4] University of Washington    [5] CIIRC CTU

megapose6d.github.io

**Abstract:** We introduce MegaPose, a method to estimate the 6D pose of novel objects, that is, objects unseen during training. At inference time, the method only assumes knowledge of (i) a region of interest displaying the object in the image and (ii) a CAD model of the observed object. The contributions of this work are threefold. First, we present a 6D pose refiner based on a render & compare strategy which can be applied to novel objects. The shape and coordinate system of the novel object are provided as inputs to the network by rendering multiple synthetic views of the object's CAD model. Second, we introduce a novel approach for coarse pose estimation which leverages a network trained to classify whether the pose error between a synthetic rendering and an observed image of the same object can be corrected by the refiner. Third, we introduce a large scale synthetic dataset of photorealistic images of thousands of objects with diverse visual and shape properties, and show that this diversity is crucial to obtain good generalization performance on novel objects. We train our approach on this large synthetic dataset and apply it *without retraining* to hundreds of novel objects in real images from several pose estimation benchmarks. Our approach achieves state-of-the-art performance on the ModelNet and YCB-Video datasets. An extensive evaluation on the 7 core datasets of the BOP challenge demonstrates that our approach achieves performance competitive with existing approaches that require access to the target objects during training. Code, dataset and trained models are available on the project page [1].

## 1  Introduction

Accurate 6D object pose estimation is essential for many robotic and augmented reality applications. Current state-of-the-art methods are learning-based [2, 3, 4, 5, 6] and require 3D models of the objects of interest at both training and test time. These methods require hours (or days) to generate synthetic data for each object and train the pose estimation model. They thus cannot be used in the context of robotic applications where the objects are only known during inference (e.g. CAD models are provided by a manufacturer or reconstructed [7]), and where rapid deployment to novel scenes and objects is key.

The goal of this work is to estimate the 6D pose of novel objects, *i.e.*, objects that are only available at *inference time* and are not known in advance during training. This problem presents the challenge of generalizing to the large variability in shape, texture, lighting conditions, and severe occlusions that can be encountered in real-world applications. Some prior works [8, 9, 10, 11, 12, 13, 14] have considered category-level pose estimation to partially address the challenge of novel objects by developing methods that can generalize to novel object instances of a known class (e.g. mugs or shoes). These methods however do not generalize to object instances outside of training categories. Other methods aim at generalizing to any novel instances regardless of their category [15, 16, 17, 18, 19, 20, 21, 22]. These works present important technical limitations. They rely on non-learning based components

---

[1] Inria Paris and Département d'informatique de l'ENS, École normale supérieure, CNRS, PSL Research University, 75005 Paris, France.

[3] LIGM, École des Ponts, Univ Gustave Eiffel, CNRS, Marne-la-vallée, France.

[5] Czech Institute of Informatics, Robotics and Cybernetics at the Czech Technical University in Prague.

[†] Work partially done while the author was an intern at NVIDIA.

6th Conference on Robot Learning (CoRL 2022), Auckland, New Zealand.

| (a) Training | (b) Inference | (c) Visually guided robotic manipulation |
|---|---|---|

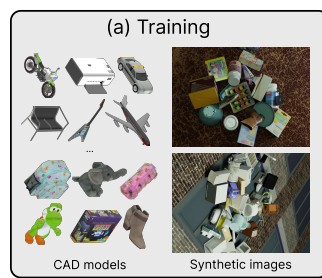
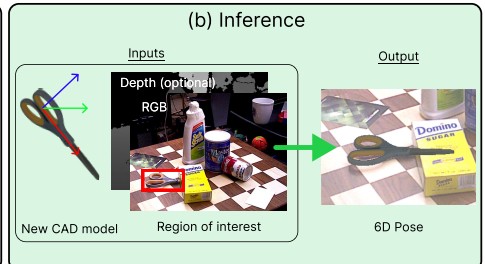
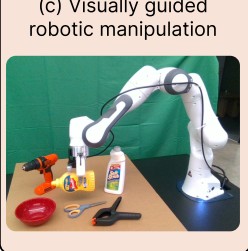

**Figure 1: MegaPose** is a 6D pose estimation approach (a) that is trained on millions of synthetic scenes with thousands of different objects and (b) can be applied *without re-training* to estimate the pose of any novel object, given a CAD model and a region of interest displaying the object. It can thus be used to rapidly deploy visually guided robotic manipulation systems in novel scenes containing novel objects (c).

for generating pose hypotheses [21] (e.g. PPF [23]), for pose refinement [17] (e.g. PnP [24] and ICP [25, 26]), for computing photometric errors in pixel space [15], or for estimating the object depth [18, 16] (e.g. using only the size of a 2D detection [27]). These components however inherently cannot benefit from being trained on large amount of data to gain robustness with respect to noise, occlusions, or object variability. Learning-based methods also have the potential to improve as the quality and size of the datasets improve.

Pipelines for 6D pose estimation of known (not novel) objects that consist of multiple learned stages [4, 5] have shown excellent performance on several benchmarks [2] with various illumination conditions, textureless objects, cluttered scenes and high levels of occlusions. We take inspiration from [4, 5] which split the problem into three parts: (i) 2D object detection, (ii) coarse pose estimation, and (iii) iterative refinement via render & compare. We aim at extending this approach to novel objects unseen at training time. The detection of novel objects has been addressed by prior works [17, 28, 29, 30] and is outside the scope of this paper. In this work, we focus on the coarse and refinement networks for 6D pose estimation. Extending the paradigm from [4] presents three major challenges. First, the pose of an object depends heavily on both its visual appearance and choice of coordinate system (defined in the CAD model of the object). In existing refinement networks based on render & compare [20, 4], this information is encoded in the network weights during training, leading to poor generalization results when tested on novel objects. Second, direct regression methods for coarse pose estimation are trained with specific losses for symmetric objects [4], requiring that object symmetries be known in advance. Finally, the diversity of shape and visual properties of the objects that can be encountered in real-world applications is immense. Generalizing to novel objects requires robustness to properties such as object symmetries, variability of object shape, and object textures (or absence of).

**Contributions.** We address these challenges and propose a method for estimating the pose of any novel object in a single RGB or RGB-D image, as illustrated in Figure 1. First, we propose a novel approach for 6D pose refinement based on render & compare which enables generalization to novel objects. The shape and coordinate system of the novel object are provided as inputs to the network by rendering multiple synthetic views of the object's CAD model. Second, we propose a novel method for coarse pose estimation which does not require knowledge of the object symmetries during training. The coarse pose estimation is formulated as a classification problem where we compare renderings of random pose hypotheses with an observed image, and predict whether the pose can be corrected by the refiner. Finally, we leverage the availability of large-scale 3D model datasets to generate a highly diverse synthetic dataset consisting of 2 million photorealistic [31] images depicting over 20K models in physically plausible configurations. The code, dataset and trained models are available on the project page [1].

We show that our novel-object pose estimation method trained on our large-scale synthetic dataset achieves state-of-the-art performance on ModelNet [32, 20]. We also perform an extensive evaluation of the approach on hundreds of novel objects from all 7 core datasets of the BOP challenge [2] and demonstrate that our approach achieves performance competitive with existing approaches that require access to the target objects during training.

## 2 Related work

In this section, we first review the literature on 6D pose estimation of known rigid objects. We then focus on the practical scenario similar to ours where the objects are not known prior to training.

**6D pose estimation of known objects**. Estimating the 6D pose of rigid objects is a fundamental computer vision problem [33, 34, 35] that was first addressed using correspondences established with locally invariant features [35, 36, 37, 38, 23] or template matching [39, 40]. These have been replaced by learning-based methods with convolutional neural networks that directly regress sets of sparse [41, 42, 27, 43, 44, 45, 46] or dense [47, 48, 49, 50, 3, 44] features. All these approaches use non-learning stages relying on PnP+Ransac [51, 24] to recover the pose from correspondences in RGB images, or variants of the iterative closest point algorithm, ICP [25, 26], when depth is available. The best performing methods rely on trainable refinement networks [52, 20, 4, 20, 5] based on render & compare [53, 54, 55, 20]. These methods render a single image of the object, which is not sufficient to provide complete information on the shape and coordinate system of a 3D model to the network. This information is thus encoded in the networks weights when training, which leads to poor generalization when tested on novel objects unseen at training. Our approach renders multiple views of an object to provide this 3D information, making the trained network independent of these object-specific properties.

**6D pose estimation of novel objects.** Other works consider a practical scenario where the objects are not known in advance. Category-level 6D pose estimation is a popular problem [8, 9, 10, 11, 12, 13, 14] in which CAD models of test objects are not known, but the objects are assumed to belong to a known category. These methods rely on object properties that are common within categories (*e.g.* handle of a mug) to define and estimate the object pose, and thus cannot generalize to novel categories. Our method requires the 3D model of the novel object instance to be known during inference, but does not rely on any category-level information. Other works address a scenario similar to ours. [56, 19, 57, 18, 16, 30] only estimate the 3D orientation of novel objects by comparing rendered pose hypotheses with the observed image using features extracted by a network. They rely on handcrafted [18, 16] or learning-based DeepIM [19] refiners to recover accurate 6D poses. We instead propose a method that estimates the full 6D pose of the object and show our refinement network significantly outperforms DeepIM [20] when tested on novel object instances. The closest works to ours are OSOP [17] and ZePHyR [21]. OSOP focuses on the coarse estimation by explicitly predicting 2D-2D correspondences between a single rendered view of the object and the observed image, and solves for the pose using PnP or Kabsch [25] which makes inference slower and less robust compared to directly predicting refinement transforms with a network as done in our solution. ZePHyR [21] strongly relies on the depth modality, whereas our approach can also be used in RGB-only images. Finally, [15, 58, 22, 59] investigate using a set of real reference views of the novel object instead of using a CAD model. These approaches have only reported results on datasets with limited or no occlusions. Our use of a deep render & compare network trained on a large-scale synthetic dataset displaying highly occluded object instances enables us to handle highly cluttered scenes with high occlusions like in the LineMOD Occlusion, HomebrewedDB or T-LESS datasets.

## 3 Method

In this section we present our framework for pose estimation of novel objects. Our goal is to detect the pose $\mathcal{T}_{CO}$ (the pose of object frame $O$ expressed in camera frame $C$ composed of 3D rotation and 3D translation) of a novel object given an input RGB (or RGBD) image, $I_o$, and a 3D model of the object. Similar to DeepIM [20] and CosyPose [4], our method consists of three components (1) object detection, (2) coarse pose estimation and (3) pose refinement. Compared to these works, our proposed method enables generalization to novel objects not seen during training, requiring novel approaches for the coarse model, the refiner and the training data. Our approach can accept either RGB or RGBD inputs, if depth is available the RGB and D images are concatenated before being passed into the network. Detection of novel-objects in an image is an interesting problem that has been addressed in prior work [28, 60, 17, 22, 30] but lies outside the scope of this paper. Thus for our experiments we assume access to an object detector, but emphasize that our method can be coupled with any object detector, including zero-shot methods such as those in [28, 60].

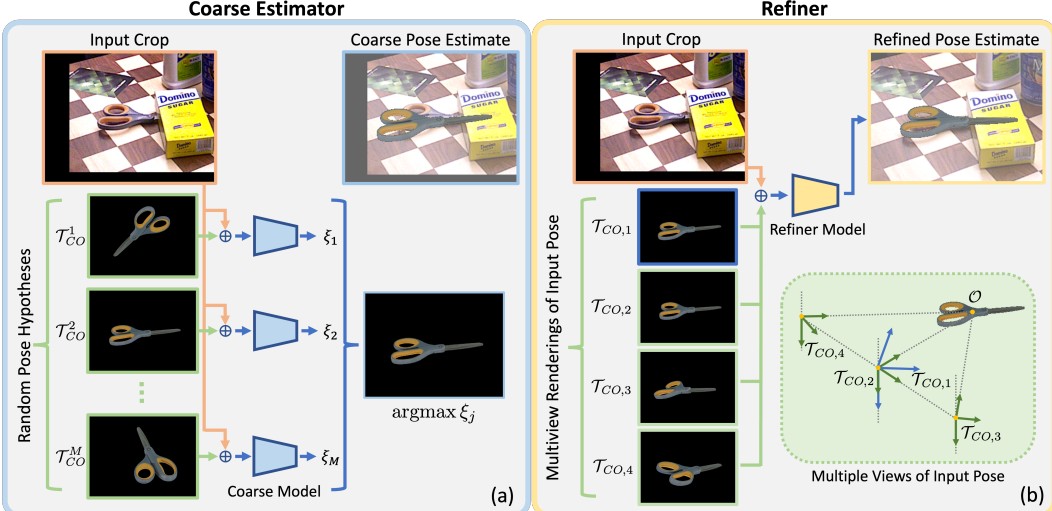

**Figure 2:** $\oplus$ denotes concatenation. **(a) Coarse Estimator:** Given a cropped input image the coarse module renders the object in multiple input poses $\{\mathcal{T}_{CO}^{j}\}$. The coarse network then classifies which rendered image best matches the observed image. **(b) Refiner:** Given an initial pose estimate $\mathcal{T}_{CO}^{k}$ the refiner renders the objects at the estimated pose $\mathcal{T}_{CO,1} := \mathcal{T}_{CO}^{k}$ (blue axes) along with 3 additional viewpoints $\{\mathcal{T}_{CO,i}\}_{i=2}^{4}$ (green axes) defined such that the camera $z$-axis intersects the anchor point $\mathcal{O}$. The refiner network consumes the concatenation of the observed and rendered images and predicts an updated pose estimate $\mathcal{T}_{CO}^{k+1}$.

## 3.1 Technical Approach

**Coarse pose estimation.** Given an object detection, shown in Figure 1(b), the goal of the coarse pose estimator is to provide an initial pose estimate $\mathcal{T}_{CO,\text{coarse}}$ which is sufficiently accurate that it can then be further improved by the refiner. In order to generalize to novel-objects we propose a novel classification based approach that compares observed and rendered images of the object in a variety of poses and selects the rendered image whose object pose best matches the observed object pose.

Figure 2(a) gives an overview of the coarse model. At inference time the network consumes the observed image $I_o$ along with rendered images $\{I_r(\mathcal{T}_{CO}^{j})\}_{j=1}^{M}$ of the object in many different poses $\{\mathcal{T}_{CO}^{j}\}_{j=1}^{M}$. For each pose $\mathcal{T}_{CO}^{j}$ the model predicts a score $(I_o, I_r(\mathcal{T}_{CO}^{j})) \rightarrow \xi_j$ that classifies whether the pose hypothesis is within the basin of attraction of the refiner. The highest scoring pose $\mathcal{T}_{CO}^{j^*}, j^* = \arg\max_j \xi_j$ is used as the initial pose for the refinement step. Since we are performing classification, our method can implicitly handle object symmetries, as multiple poses can be classified as correct.

**Pose refinement model.** Given an input image and an estimated pose, the refiner predicts an updated pose estimate. Starting from a coarse initial pose estimate $\mathcal{T}_{CO,\text{coarse}}$ we can iteratively apply the refiner to produce an improved pose estimate. Similar to [4, 20] our refiner takes as input observed $I_o$ and rendered images $I_r(\mathcal{T}_{CO}^{k})$ and predicts an updated pose estimate $\mathcal{T}_{CO}^{k+1}$, see Figure 2 (b), where $k$ refers to the $k^{th}$ iteration of the refiner. Our pose update uses the same parameterization as DeepIM [20] and CosyPose [4] which disentangles rotation and translation prediction. Crucially this pose update $\Delta\mathcal{T}$ depends on the choice of an *anchor point* $\mathcal{O}$, see Appendix for more details. In prior work [4, 20] which trains and tests on the same set of objects, the network can effectively learn the position of the anchor point $\mathcal{O}$ for each object. However in order to generalize to novel objects we must enable the network to infer the anchor point $\mathcal{O}$ at inference time.

In order to provide information about the anchor point to the network we always render images $I_r(\mathcal{T}_{CO}^{k})$ such that the anchor point $\mathcal{O}$ projects to the image center. Using rendered images from multiple distinct viewpoints $\{\mathcal{T}_{CO,i}\}_{i=1}^{N}$ the network can infer the location of the anchor point $\mathcal{O}$ as the intersection point of camera rays that pass through the image center, see Figure 2(b).

Additional information about object shape and geometry can be provided to the network by rendering depth and surface normal channels in the rendered image $I_r$. We normalize both input depth

(if available) and rendered depth images using the currently estimated pose $\mathcal{T}_{CO}^k$ to assist the network in generalizing across object scales, see Appendix for more details.

**Network architecture.** Both the coarse and refiner networks consists of a ResNet-34 backbone followed by spatial average pooling. The coarse model has a single fully-connected layer that consumes the backbone feature and outputs a classification logit. The refiner network has a single fully-connected layer that consumes the backbone feature and outputs 9 values that specify the translation and rotation for the pose update.

### 3.2 Training Procedure

**Training data.** For training, both the coarse and refiner models require RGB(-D)[1] images with ground-truth 6D object pose annotations, along with 3D models for these objects. In order for our approach to generalize to novel-objects we require a large dataset containing diverse objects. All of of our methods are trained purely on synthetic data generated using BlenderProc [31]. We generate a dataset of 2 million images using a combination of ShapeNet [61] (abbreviated as SN) and Google-Scanned-Objects (abbreviated as GSO) [7]. Similar to the BOP [62] synthetic data, we randomly sampled objects from our dataset and dropped them on a plane using a physics simulator. Materials, background textures, lighting and camera positions are randomized. Example images can be seen in Figure 1(a) and in the Appendix. Some of our ablations also use the synthetic training datasets provided by the BOP challenge [62]. We add data augmentation similar to CosyPose [4] to the RGB images which was shown to be a key to successful sim-to-real transfer. We also apply data augmentation to the depth images as explained in the appendix.

**Refiner model.** The refiner model is trained similarly to [4]. Given an image with an object $\mathcal{M}$ at ground-truth pose $\mathcal{T}_{CO}^*$ we generate a perturbed pose $\mathcal{T}_{CO}'$ by applying a random translation and rotation to $\mathcal{T}_{CO}^*$. Translation is sampled from a normal distribution with a standard deviations of $(0.02, 0.02, 0.05)$ centimeters and rotation is sampled as random Euler angles with a standard deviation of $15$ degrees in each axis. The network is trained to predict the relative transformation between the initial and target pose. Following [4, 20] we use a loss that disentangles the prediction of depth, $x$-$y$ translation, and rotation. See the appendix for more details.

**Coarse model.** Given an input image $I_o$ of an object $\mathcal{M}$ and a pose $\mathcal{T}_{CO}'$ the coarse model is trained to classify whether pose $\mathcal{T}_{CO}'$ is within the basin of attraction of the refiner. In other words, if the refiner were started with the initial pose estimate $\mathcal{T}_{CO}'$ would it be able to estimate the ground-truth pose via iterative refinement? Given a ground-truth pose-annotation $\mathcal{T}_{CO}^*$ we randomly sample poses $\mathcal{T}_{CO}'$ by adding random translation and rotation to $\mathcal{T}_{CO}^*$. The positives are sampled from the same distribution used to generate the perturbed poses the refiner network is trained to correct (see above), and other poses sufficiently distinct to this one (see the appendix for more details) are marked as negatives. The model is then trained with binary cross entropy loss.

## 4 Experiments

We evaluate our method for 6D pose estimation of novel objects using the seven challenging datasets of the BOP [2, 62] 6D pose estimation benchmark, and the ModelNet [20] dataset. The dataset and the standard 6D pose estimation metrics we use are detailed in Section 4.1. In all our experiments, the objects are considered novel, i.e. they are only available during inference on a new image and they are not used during training. In Section 4.2, we evaluate the performance of our approach composed of coarse and refinement networks. Notably, we show that (i) our method is competitive with others that require the object models to be known in advance, and (iii) our refiner outperforms current state-of-the-art on the ModelNet and YCB-V datasets. Section 4.3 validates our technical contributions and shows the crucial importance of the training data in the success of our method. Finally, we discuss the limitations in Section 4.4.

---

[1]Our method can consume either RGB or RGB-D images depending on the input modalities that are available.

| Pose Initialization | | Pose Refinement | | | BOP Datasets | | | | | | | |
|---|---|---|---|---|---|---|---|---|---|---|---|---|
| Method | Novel objects | Method | Novel objects | RGB-D Input | LM-O | T-LESS | TUD-L | IC-BIN | ITODD | HB | YCB-V | Mean |
| 1 CosyPose [4] | ✗ | CosyPose | ✗ | | 63.3 | 64.0 | 68.5 | 58.3 | 21.6 | 65.6 | 57.4 | 57.0 |
| 2 SurfEmb [3] | ✗ | BFGS | ✗ | | 66.3 | 73.5 | 71.5 | 58.8 | 41.3 | 79.1 | 64.7 | 65.0 |
| 3 SurfEmb [3] | ✗ | BFGS+ICP | ✓ | ✓ | *75.8* | *82.8* | *85.4* | *65.6* | 49.8 | *86.7* | *80.6* | *75.2* |
| 4 OSOP [17] | ✓ | Multi-Hyp. | ✓ | | 31.2 | - | - | - | - | 49.2 | 33.2 | - |
| 5 OSOP [17] | ✓ | MH+ICP | ✓ | ✓ | 48.2 | - | - | - | - | 60.5 | 57.2 | - |
| 6 (PPF, Sift) + Zephyr [21] | ✓ | - | ✓ | ✓ | **59.8** | - | - | - | - | - | 51.6 | - |
| 7 (PPF, Sift) + Our coarse | ✓ | Our refiner | ✓ | ✓ | 57.0 | - | - | - | - | - | 62.3 | - |
| 8 CosyPose [4] | ✗ | – | | | 53.6 | 52.0 | 57.6 | 53.0 | 13.1 | 33.5 | 33.3 | 42.3 |
| 9 CosyPose [4] | ✗ | Ours | ✓ | | 65.5 | 72.0 | 70.1 | 57.3 | 28.4 | 67.0 | 56.8 | 59.6 |
| 10 CosyPose [4] | ✗ | Ours | ✓ | ✓ | 71.2 | 63.8 | 85.0 | 55.1 | 39.9 | 73.2 | 69.2 | 66.0 |
| 11 Ours | ✓ | – | | | 18.7 | 19.7 | 20.5 | 15.3 | 8.00 | 18.6 | 13.9 | 16.2 |
| 12 Ours | ✓ | Ours | ✓ | | 53.7 | **62.2** | 58.4 | **43.6** | 30.1 | 72.9 | 60.4 | 54.5 |
| 13 Ours | ✓ | Ours | ✓ | ✓ | 58.3 | 54.3 | **71.2** | 37.1 | **40.4** | **75.7** | **63.3** | **57.2** |

**Table 1: Results on the BOP challenge datasets.** We report the AR score on each of the 7 datasets considered in the BOP challenge and the mean score across datasets. With the exception of Zephyr (row 11), all approaches are trained purely on synthetic data. For each column, we denote the best over result in *italics* and the best novel-object pose estimation method in **bold**.

## 4.1 Dataset and metrics

We consider the seven core datasets of the BOP challenge [62, 2]: LineMod Occlusion (LM-O) [63], T-LESS [64], TUD-L [62], IC-BIN [65], ITODD [66], HomebrewedDB (HB) [67] and YCB-Video (YCB-V) [47]. These datasets exhibit 132 different objects in cluttered scenes with occlusions. These objects present many factors of variation: textured or untextured, symmetric or asymmetric, household or industrial (e.g. watcher pitcher, stapler, bowls, multi-socket plug adaptor) which makes them representative of objects that are typically encountered in robotic scenarios. The ModelNet dataset depicts individual instances of objects from 7 classes of the ModelNet [32] dataset (bathtub, bookshelf, guitar, range hood, sofa, tv stand and wardrobe). We use initial poses provided by adding noise to the ground truth, similar to previous works [20, 16, 15]. The focus is on refining these inital poses. We follow the evaluation protocol of [2] for BOP datasets, and of DeepIM [20] for ModelNet.

## 4.2 6D pose estimation of novel objects

**Performance of coarse+refiner.** Table 1 reports results of our novel-object pose estimation method on the BOP datasets. We first use the detections and pose hypotheses provided by a combination of PPF and SIFT, similar to the state-of-the-art method Zephyr [21]. For each object detection, these algorithms provide multiple pose hypotheses. We find the best hypothesis using the score of our coarse network, and apply 5 iterations of our refiner. Results are reported in row 7. On YCB-V, our method achieves a +10.7 AR score improvement compared to Zephyr (row 6). Averaged across the YCB-V and LM-O datasets, the AR score of our approach is 59.7 compared to 55.7 for Zephyr (row 6). Next, we provide a complete set of results using the detections from Mask-RCNN networks. Please note that since detection of novel objects is outside the scope of this paper, we use the networks trained on the synthetic PBR data of the target objects [2] which are publicly available for each dataset. We report the results of our coarse estimation strategy (Table 1, row 11), and after running the refiner network, on RGB (row 12) or RGB-D (row 13) images. We observe that (i) our refinement network significantly improves the coarse estimates (+41.0 mean AR score for our RGBD refiner) and (ii) the performance of both models is competitive with the learning-based refiner of CosyPose [4] (row 1) while not requiring to be trained on the test objects. The recent SurfEmb [3] performs better than our approach, but heavily relies on the knowledge of the objects for training and cannot generalize to novel objects.

**Performance of the refiner.** We now focus on the evaluation of our refiner which can be used to refine arbitrary initial poses. Our refiner is the only learning-based approach in Table 1 which can be applied to novel objects. In rows 9 and 10, we apply our refiner to the coarse estimates of CosyPose [4] (row 11). Again, we observe that our refiner significantly improves the accuracy of these initial pose estimates (+23.7 in average for the RGB-D model). Notably, the RGB-only method (row 9) performs better than the CosyPose refiner (row 1) on average, while not having seen the BOP objects during

| Method | RGB-D | Average Recall | | |
| --- | --- | --- | --- | --- |
| | | (5°, 5cm) | ADD (0.1d) | Proj2D (5px) |
| DeepIM [20] | ✓ | 64.3 | 83.6 | 73.3 |
| Multi-Path [16] | | 84.8 | 90.1 | 81.6 |
| LatentFusion[15] | ✓ | 85.5 | 94.3 | 94.7 |
| Ours | | 88.6 | 90.5 | 88.9 |
| Ours | ✓ | **97.6** | **98.9** | **97.5** |

**Table 2: Evaluation of the refiner on the ModelNet [20] dataset.** The mean average recall is computed over the seven classes of the dataset.

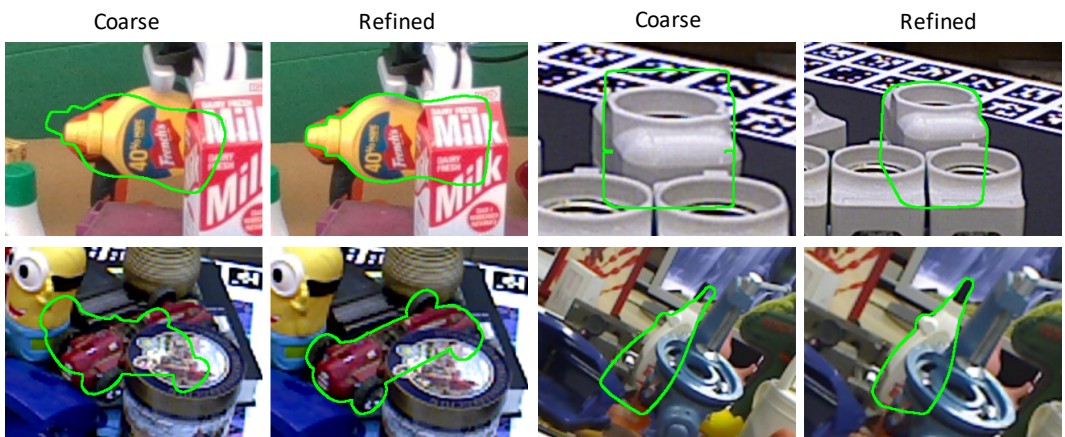

**Figure 3: Qualitative results.** For each pair of images, the left image is a visualization of our coarse estimate, and the right image is after applying 5 iterations of our refiner. None of these objects from the YCBV, LMO, HB, or T-LESS datasets were used for training our approach. Please notice the high accuracy of MegaPose despite (i) severe occlusions and (ii) the varying properties of the novel objects (e.g. the texture-less industrial plug in the top-right example, textured mustard bottle in the top-left).

training. This is thanks to our large-scale training on thousands of various objects, while CosyPose is only trained on tens of objects for each dataset.

One iteration of our refiner takes approximately 50 milliseconds on a RTX 2080 GPU, making it suitable for use in an online tracking application. Five iterations of our refiner are also 5 times faster than the object-specific refiner of SurfEmb [3] which takes around 1 second per image crop. Finally, we evaluate our refiner on ModelNet and compare it with the state-of-the-art methods MP-AAE [16] and LatentFusion [15]. For this experiment, we remove the ShapeNet categories that overlap with the test ones in ModelNet from our training set in order to provide a fair comparison on novel instances and novel categories similar to [15, 16, 20]. Results reported in Table 2 show that our refiner significantly outperforms existing approaches across all metrics.

### 4.3 Ablations

In this section we perform ablations of our approach to validate our main contributions. Additional ablations are in the appendix. For these ablations, we consider the RGB-only refiner and re-train several models with different configurations of hyper-parameters and training data.

**Encoding the anchor point and object shape.** As discussed in Section 3.1 the refiner must have information about the anchor point $\mathcal{O}$ in order to generalize to novel objects. We accomplish this by using 4 rendered views pointing towards the anchor point, see Figure 2(b). Table 3(a) shows the performance of the refiner increases as we increase the number of views from 1 to 4, validating our design choice. Multiple views may also help the network to understand the object's appearance from alternate viewpoints, thus potentially helping the refiner to overcome large initial pose errors. We also validate our choice to provide a normal map of the object to the network. This information can help the network use subtle object appearance variations that are only visible under different illumination like the details on the cross of a guitar.

| Rendered views | Rendered normals | BOP5 | ModelNet ADD(0.1d) |
|---|---|---|---|
| 1 | ✓ | 52.0 | 83.3 |
| 2 | ✓ | 59.0 | 90.4 |
| 4 | ✓ | **61.7** | **96.1** |
| 4 |  | 59.1 | 83.1 |

(a)

| Training objects | Num. objects | BOP5 | ModelNet ADD(0.1d) |
|---|---|---|---|
| GSO+ShapeNet | 10 + 100 | 47.9 | 28.7 |
| GSO+ShapeNet | 100 + 1000 | 49.3 | 80.3 |
| GSO+ShapeNet | 250 + 2000 | 56.9 | 82.0 |
| GSO+ShapeNet | 500 + 10000 | 59.3 | 89.3 |
| GSO+ShapeNet | 1000 + 20000 | 61.7 | **96.1** |
| GSO | 1000 | 62.2 | 95.7 |
| BOP | 132 | **62.6** | 93.4 |

(b)

**Table 3: Ablation study.** We study (a) using multiple rendered object views and normal maps as input to our RGB-only refiner model and (b) training the refiner on different variations of our large-scale synthetic dataset. Average recall is reported on BOP5 (mean of LM-O, T-LESS, TUD-L, IC-BIN and YCB-V) and ModelNet.

**Number of training objects.** We now show the crucial role of the training data. We report in Table 3 (b) the results for our refiner trained on an increasing number of CAD models. The performance steadily increases with the number of objects, which validates that training on a large number of object models is important to generalize to novel ones. These results also suggest that the performance of our approach *could* be improved as more datasets of high-quality CAD models like GSO [7] become available.

**Variety in the training objects.** Next, we restrict the training to different sets of objects. We observe in the bottom of Table 3(b) that models from the GoogleScannedObjects are more important to the performance of the method on the BOP dataset compared to using both ShapeNet and GSO. We hypothesize this is due to the presence of high-quality textured objects in the GSO dataset. Finally, we train our model on the 132 objects of the BOP dataset. When testing on the same BOP objects, the performance benefits from knowing these objects during training is small compared to using our GSO+ShapeNet or GSO dataset.

## 4.4 Limitations

While MegaPose shows promising results in robot experiments (**please see the supplementary video**) and 6D pose estimation benchmarks, there is still room for improvement. We illustrate the failure modes of our approach in the supplementary material. The most common failure mode is due to inaccurate initial pose estimates from the coarse model. The refiner model is trained to correct poses that are within a constrained range but can fail if the initial error is too large. There exist multiple potential approaches to alleviate this problem. We can increase the number of pose hypotheses $M$ at the expense of increased inference time, improve the accuracy of the coarse model, and increase the basin of attraction for refinement model. Another limitation is the runtime of our coarse model. We use $M = 520$ pose hypotheses per object which takes around 2.5 seconds to be rendered and evaluated by our coarse model. In a tracking scenario however, the coarse model is run just once at the initial frame and the object can be tracked using the refiner which runs at 20Hz. Additionally, our refiner could also be coupled with alternate coarse estimation approaches such as [17, 18] to achieve improved runtime performance.

## 5 Conclusion

We propose MegaPose, a method for 6D pose estimation of novel objects. Megapose can estimate the 6D pose of novel objects given a CAD model of the object available only at test time. We quantitatively evaluated MegaPose on hundreds of different objects depicted in cluttered scenes, and performed ablation studies to validate our network design choices and highlight the importance of the training data. We will release our models and large-scale synthetic dataset to stimulate the development of novel methods that are practical to use in the context of robotic manipulation where rapid deployment to new scenes with new objects is crucial. While this work focuses on the coarse estimation and fine refinement of an object pose, detecting any unknown object given only a CAD model is still a difficult problem that remains to be solved for having a complete framework for detection and pose estimation of novel objects. Future work will address zero-shot object detection using our large-scale synthetic dataset.

## Acknowledgements

This work was partially supported by the HPC resources from GENCI-IDRIS (Grant 011011181R2), the European Regional Development Fund under the project IMPACT (reg. no. CZ.02.1.01/0.0/0.0/15 003/0000468), EU Horizon Europe Programme under the project AGIMUS (No. 101070165), Louis Vuitton ENS Chair on Artificial Intelligence, and the French government under management of Agence Nationale de la Recherche as part of the "Investissements d'avenir" program, reference ANR-19-P3IA-0001 (PRAIRIE 3IA Institute).

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
