# OpenReview forum: "MegaPose: 6D Pose Estimation of Novel Objects via Render & Compare"
_robot-learning.org/CoRL/2022/Conference — CoRL 2022 Poster_

### Official Review · Reviewer_sFyM · 2022-07-20

**Originality:** Very Good
**Technical Quality:** Very Good
**Clarity Of Presentation:** Very Good
**Impact:** 4

**Recommendation:**

Weak Accept: I recommend accepting the paper, but will not argue for my recommendation if the majority of other reviewers have a different opinion.

**Summary:**

A method is proposed with which the pose of novel/unseen objects are estimated without requiring CAD models at test time, in contrast to the majority of methods that train with those models. During inference, the input is a region of interest (from a detector) as well as the CAD model from which a coarse pose is estimated then refined using two separate networks. The coarse pose estimation network is a binary classifier indicating if the synthetic rendering of the object can be successfully refined w.r.t. the observation. The input from a set of renderings with highest score is then the initial pose to start the refiner. The refiner network regresses the translation and rotation to iteratively bring the model into alignment with the observation. Experiments on the BOP core datasets as well as ModelNet show the proposed approach achieves best performance for novel objects and competitive results compared to methods that know the object models apriori i.e. seen objects.

**Issues:**

The inability to train the coarse model that regresses instead of classifies is speculated to be caused by symmetries. Could symmetric objects be removed from the dataset, and the model retrained, to support this hypothesis?

The authors should state what hardware is used to train the models and how long training time is.

It would be good to see some qualitative failure cases.

It is not clear how the grasp poses are determined for the grasping demonstrations in the supplementary.

**Quality Of The Limitations Section:**

Limitations are addressed clearly

**Reviewer Expertise:**

4: The reviewer is confident but not absolutely certain that the evaluation is correct

**Robotics Focus:**

Highly relevant to robotics but no hardware experiments

**Strengths And Weaknesses:**

**STRENGTHS**

The paper addresses a relevant problem in the robotics community and sufficiently motivates the design choices in the context of real limitations i.e. not always having access to object models before deployment. The authors cover the literature well, enabling them to position their work in the context of prior work.

Estimating the pose of unseen objects is very challenging. Currently there exist some solutions but they have limitations that are addressed in the proposed approach. Therefore, this paper offers novelty in terms of problem formulation and methodology.

The proposed approach achieves state-of-the-art performance on BOP and ModelNet in comparison to methods that also deal with unseen objects. Additionally, robotic grasping experiments are showcased in the supplementary video to demonstrate the real-world application.


**WEAKNESSES**

While the presented solution is fully-learned, I do not see this is a critical argument as to why the method is better than those discusses on lines 36-41. It is not a strong argument to say that non-learned components are unfavourable because they depend on hyperparameters, especially when the presented work also has its own set of tunable parameters e.g. M for the number of poses to render, k for the number of iterations and even the range of sampled standard deviations for the refiner dataset. A side not to this is that the method seems to be very sensitive to M, here large M means potentially better initial poses at the cost of computation time. It would be good to see an experiment to evaluate this trade-off to complement the statements made in Sec 4.4.

I question whether the method itself contributes to the high accuracy or if a significant performance gain is attributed to the enormous dataset used to train the networks. As table 3(b) reveals, fewer object models lead to a performance drop and under 500 + 10000, the method no longer outperforms the competitors. A better experiment would be to control for the dataset (i.e. train all methods with the new data or an equivalent number of training samples) in order to fairly compare the methods. The authors themselves make the claim that their method performs well because of the dataset in comparison to others that have a limited dataset to train in (lines 227-229).

While it is ok to declare the detection stage to be out of scope (and sufficiently handled by prior work), the experiments do not represent a realistic scenario of a fully unseen object pipeline since the detections (for BOP) come from a network trained on the relevant objects, hence they are known before. A more suitable method, in the context of the paper's goals, would be one of those mentioned on line 124. I suspect that the method could be sensitive to the initial detections, thus, it would be interesting to see an ablation on how the method performs w.r.t. detection inaccuracies. If the inaccuracy of zero-shot methods is below the tolerance of the proposed method, then a full pipeline is in fact not yet achieved. Lastly, it is not clear where the detections come from for the ModelNet experiments and the demonstrations in the supplementary video.

Table 1 is misleading since so few results are provided for the most relevant comparisons (rows 4-7). One can see that PPF + Zephyr + ICP is the best on LM-O, thus, it is questionable if the proposed method is the best overall without the missing results. (I understand the authors do not provide results on other datasets but code for their method is provided, thus it seems feasible to generate results for this publication). Furthermore, the authors state it is difficult to directly compare to OSOP due to the lack of a detector. Is it possible to use their detections to initialise your approach? This would make the comparison much more fair as the masks generated by Mask RCNN trained on the known objects is a huge advantage.

Finally, the demonstration of robotic grasping in the supplementary video is a good addition but it does not constitute a real "experiment" that evaluates the proposed method's performance. No details are provided about these demonstrations so it is difficult to draw any serious conclusions.

**Summary Of Recommendation:**

The authors have presented very interesting ideas that are novel for the purpose of unseen object pose estimation/refinement. Additionally, the method outperforms current SotA approaches with similar constraints. While the submission has some very solid strengths, unfortunately the evaluation has some flaws. Namely, there is a disparity in terms of the size of the dataset methods are trained on and the detections that are used for initialisation. Also, the most related methods are not evaluated on all datasets, which presents an incomplete picture. Thus, I consider some work still be done to fully convince readers of the practical significance of the work, which leads me to recommend a weak accept.

---

> ### Author Response · Authors · 2022-08-25
> **Thank you for your review, we respond to individual comments.**
>
> [1/2]
>
> We thank the reviewer for their feedback and provide answers to their questions below. Please note our response is split into two comments due to the character limit for a single comment.
>
> ## Motivation for a learning-based approach
> We agree with the reviewer that learning-based methods also depend on hyper-parameters. We will update the sentence in lines 37-40 to motivate the use of a learning-based approach: (i) for pose estimation of known objects, learning-based methods have outperformed non-learning methods (see for examples results in the BOP Challenge 2020 [1]), and (ii) learning-based methods have the potential to improve as the quality/size of the datasets improves, something that hand-designed methods cannot take advantage of.
>
> ## Number of coarse pose hypotheses M
> We ran an additional experiment to evaluate the trade-off between performance and computation time for the number $M$ of initial hypotheses. The performance significantly improves from M=104 to M=520 (+11.4 AR on BOP5) while keeping the running-time of the coarse model reasonable (1.6 seconds for M=520 compared to 0.3 seconds for M=120). Above M=520, the performance improvement is marginal, e.g. (+0.9 AR) for 4608 hypotheses. Please note that we are still making improvements to our code and have lower runtimes than reported in the paper (1.6s for M=520 compared to the 4s mentioned in line 276).
>
> ## Validation of our contributions
> The dataset, which we presented as one of the contributions of the paper, is indeed important to the success of our learning-based method as demonstrated in the ablation presented in Table 3 (b). We will release our dataset to spur other researchers to build off this work and develop learning-based methods that can leverage this large-scale synthetic dataset. The other technical contributions of our method―(i) using multiple views of the novel object as input to the refinement network to provide information about the anchor point and object’s shape and (ii) our classification-based coarse model―are evaluated in Section 4.3 of the initial submission. These ablations validate our technical contributions independently of the data being used.
>
>
> ## Fully unseen object pipeline for detection + pose estimation
> As suggested by the reviewer, we have carried out an additional experiment to demonstrate our method can be used in a full pipeline for unseen object pose estimation. First, we replace our detector and pose hypothesis generation strategy by the same pose hypotheses used in Zephyr [20]. These are generated using a point-pair features algorithm + ICP from Drost et. al. [22] along with hypotheses from Dense SIFT feature matching. We then apply our coarse scoring network to find the best hypothesis, and run our refiner starting from this best hypothesis (as determined by our coarse model). The AR results are shown in the table below.
>
> |                                             | LM-O     | YCB-V    | Mean     |
> |---------------------------------------------|----------|----------|----------|
> | (Drost,PPF+Sift)+Zephyr+ICP                 | **59.8** | 51.6     | 55.7     |
> | (Drost,PPF+Sift)+Our coarse+Our RGB refiner | 57.0     | **62.3** | **59.7** |
>
> PPF [22] + Zephyr [20] + ICP (line 7 in Table 1 in the submission) is the best on LM-O (+2.8 compared to ours), but is significantly worse on YCB-V (-10.7). Averaged over these two datasets, our pose estimation approach (59.7) significantly outperforms Zephyr + ICP (55.7). Generating hypotheses using PPF+Sift does not require any training and can be applied to novel objects. These results thus demonstrate the high performance of our pose estimation approach when it is used in a full pipeline for detection + pose estimation of novel objects.
>
> Please also note that Zephyr uses real data for training, while our approach is only trained using synthetic data. Finally, a significant advantage of our approach over Zephyr is that it can be applied to RGB images whereas Zephyr strongly relies on the depth modality as mentioned in line 106. These results demonstrate the high performance of our method in a pipeline where the target objects are not used for training a detector.
>
> ## Comparison with Zephyr
> The results presented in our response “Fully unseen object pipeline for detection + pose estimation” show that our approach significantly outperforms Zephyr on YCB-V (+10.7 AR score). Another significant advantage of our approach over Zephyr is that it can be run on RGB images, whereas Zephyr strongly relies on the depth modality, as mentioned in line 106 of our submission.
>
> ## Comparison with OSOP
> We contacted the authors of OSOP to get access to their detections and carry out a fair comparison with their approach but did not receive a response at the time of writing this rebuttal.

---

> > ### Author Response · Authors · 2022-08-25
> > **Thank you for your review, we respond to individual comments.**
> >
> > [2/2]
> >
> > ## Requirement for object masks
> > Please note that our approach only requires a 2D detection as input, and **does not** use the object masks predicted by Mask-RCNN.
> >
> > ## Detections for ModelNet and supplementary video
> > In the ModelNet experiments, the initial pose is provided and the focus is on refinement, as explained in lines 206-207. Thus, all methods in Table 2 do not rely on external detectors. In the supplementary video experiments, we use the Mask-RCNN models of CosyPose [3] which are trained on the target objects. The goal in the supplementary video is to show the quality of the poses predicted with our approach for novel objects.
> >
> > ## Robotic Grasping Experiments
> > We performed qualitative real-robot grasping experiments. For multiple YCB-V objects, we manually annotated one grasp with respect to the object’s coordinate frame. We then placed the considered object (e.g. the drill in the supplementary video) in a scene among other objects representing visual distractors. The object may be placed on the table or on another object. We then take a single RGB image of the scene using a RealSense D415 camera mounted on the gripper of a Franka Emika Panda robot. We detect the object in 2D using the Mask-RCNN detector from CosyPose [3], and run our Megapose approach composed of coarse and refiner modules for estimating the 6D pose of the object with respect to the camera. We then express the 6D pose of the object and grasp with respect to the robot using the known camera-to-robot extrinsic calibration. We then use a motion planner to generate a robot motion that reaches the estimated grasp pose with the gripper, and lift the object. The purpose of this experiment is to qualitatively show that the pose estimates are of sufficiently high quality that they are useful for a robotic manipulation task.
> > We will add these explanations in the paper, as well as additional examples in the camera-ready version of the video.
> >
> > ## Qualitative Failure Cases
> > We present qualitative examples of failure cases and discuss the main failure modes of our approach in Section 2 of the attached PDF. Please also see the response "Failure modes and performance on specific types of objects" to the reviewer KfxR.
> >
> > ## GPU Hardware and Training Time
> > We will add details on this for the camera-ready version. Training time is respectively 32 and 48 hours for the coarse and refiner models using 32 V-100 GPUs.
> >
> > ## Further investigation on the failure of the regression-based coarse model
> > We thank the reviewer for the suggestion. However, it is not possible to remove symmetric objects from our large-scale training dataset of objects because objects are not annotated with their symmetries. Our method does not rely on such annotations, which allows us to use a large number of CAD models without requirements for costly manual annotations of the symmetries.

---

> > > ### Comment · Reviewer_sFyM · 2022-08-26
> > > **Response to authors' answers [2/2]**
> > >
> > > __Requirement for object masks__ Ok, there must have been a misunderstanding here.
> > >
> > > __Detections for ModelNet and supplementary video__ What do you mean by "the initial pose is provided"?
> > >
> > > __Robotic Grasping Experiments__, __Qualitative Failure Cases__, __GPU Hardware and Training Time__ All addressed with good explanations and detailed information.
> > >
> > > __Further investigation on the failure of the regression-based coarse model__ Since you cannot experimentally support the original claim in the paper, I suggest to remove it because there is no strong evidence.

---

> > > > ### Author Response · Authors · 2022-08-27
> > > > **Response to the reviewers' additional comments**
> > > >
> > > > **What do you mean by "the initial pose is provided"?**
> > > >
> > > > In the ModelNet dataset, a set of initial poses obtained by adding noise to the ground truth are used as input to the refinement network. The poses obtained after refinement are evaluated against the ground truth. This dataset was introduced in the DeepIM paper [19], which we refer the reviewer to for more details.
> > > >
> > > > **Since you cannot experimentally support the original claim in the paper, I suggest to remove it because there is no strong evidence.**
> > > >
> > > > We agree with the reviewer that the proposed experiment is not sufficient to validate the hypothesis that the failure of our regression-based coarse baseline is only attributed to the presence of symmetric objects in our training set. Please note that we mention this as an hypothesis and not a claim (see lines 254-256). Nonetheless, we believe that mentioning that this evident baseline fails could be useful to other researchers in the field. Unless the reviewer thinks this paragraph should entirely be removed from the paper, we suggest to move the paragraph of lines 250-256 to the supplementary material, and mention that the failure could also be attributed to other factors, such as the difficulty to interpret the full 3D geometry of an object with a CNN given six views of its 3D model captured under distant viewpoints.

---

> > ### Comment · Reviewer_sFyM · 2022-08-26
> > **Response to authors' answers [1/2]**
> >
> > __Motivation for a learning-based approach__, __Number of coarse pose hypotheses M__ Thanks for the stronger arguments and additional insight with the experiments. It is recommended to make these modification and add the results to the manuscript.
> >
> > __Validation of our contributions__ I can see that section 4.3 validates the contributions independently of the data being used. However, the comparison to the state of the art still does not seem fair since other methods are trained on a different (and smaller) dataset than the presented approach. Therefore, claims about the proposed method achieving the best results is not fully attributed to the technical innovations but partly due to the data used.
> >
> > __Fully unseen object pipeline for detection + pose estimation__, __Comparison with Zephyr__ This is a very good experiment and shows a full pipeline can be achieved. A minor comment is that it seems your approach is the RGB pipeline (i.e., "Our RGB refiner"). If this is supposed to be compared to line 12 in Table 1, the initial hypotheses from (Drost, PPF+SIFT) results in better performance than starting from the Mask RCNN detections. Do you have any thoughts as to why this is the case? It would mean that the performance of all your methods (lines 11-13 in Table 1) would improve as well using the new initialisation.
> >
> > __Comparison with OSOP__ Fair enough.

---

> > > ### Author Response · Authors · 2022-08-27
> > > **Response to the reviewers' additional comments**
> > >
> > > **It is recommended to make these modification and add the results to the manuscript.**
> > >
> > > We would like to thank again the reviewer for the additionnal comments. This discussion is helping us improve the quality of the paper, and we will take into account all the feedback from the reviewers in the final version of the paper.
> > >
> > > **However, the comparison to the state of the art still does not seem fair since other methods are trained on a different (and smaller) dataset than the presented approach. Therefore, claims about the proposed method achieving the best results is not fully attributed to the technical innovations but partly due to the data used.**
> > >
> > > We agree with the reviewer that the high performance of our approach is attributed to both our contributions: the technical innovations, and the training data being used.
> > >
> > > Regarding our comparison with the state-of-the-art on ModelNet, we agree that using different training data makes it difficult to separate the influence of the data from the network architecture in this experiment. In terms of the data being used, we do not believe our comparison with the second-best method, LatentFusion, is unfair. The LatentFusion paper mentions they use all the meshes of ShapeNetCore during training with the exclusion of meshes belonging to the testing classes. This corresponds to 46336 different object models, while our method “only” uses 20000 shapenet meshes (we also exclude the testing classes as mentionned in lines 233-235) due to a memory limitation in our current implementation of the training. Finally, please note that LatentFusion requires object masks as input, and only present results on datasets with moderate (LineMOD) or no occlusions (MOPED, ModelNet). Our method does not require object masks as input, and we present quantitative results on the challenging T-LESS, IC-BIN, HomebrewedDB and LineMOD occlusion (please note this is a different, more challenging dataset than LineMOD) datasets which depict severe occlusions.
> > >
> > > **If this is supposed to be compared to line 12 in Table 1, the initial hypotheses from (Drost, PPF+SIFT) results in better performance than starting from the Mask RCNN detections. Do you have any thoughts as to why this is the case?**
> > >
> > > This difference could be attributed to Drost,PPF having access to the depth for pose hypothesis generation while the MaskRCNN model only uses rgb as input.

---

### Official Review · Reviewer_KfxR · 2022-08-01

**Originality:** Very Good
**Technical Quality:** Good
**Clarity Of Presentation:** Excellent
**Impact:** 4

**Recommendation:**

Weak Accept: I recommend accepting the paper, but will not argue for my recommendation if the majority of other reviewers have a different opinion.

**Summary:**

This manuscript presented a method to estimate the 6D pose of objects unseen during training but only requires a CAD model at testing. It is achieved by learning two key components. A pose hypothesis generator and classification that initializes some pose guesses and whether they can be handled by the refiner, the second component. The refiner follows the render-and-compare idea similar to the existing CAD-model based methods with a difference that it can refine a novel object unseen during training. A large amount of training data is generated to gain such inductive bias.

**Issues:**

- In figure 2 bottom right, what does the green and blue axis mean and why is the T_CO2 and T_CO1 overlapped (redundant)?
- During testing for pose hypothesis generation, what if the object’s frame origin defined in the CAD model is far away from the object’s geometry center, and thus far away from the 2D bounding box center? Is the current initializing process still able to converge?


**Quality Of The Limitations Section:**

Additional details required

**Reviewer Expertise:**

5: The reviewer is absolutely certain that the evaluation is correct and very familiar with the relevant literature

**Robotics Focus:**

Sufficient demonstration on hardware

**Strengths And Weaknesses:**

Strengths:
- The paper is well written and the figures are presented nicely
- The motivation is clear and the application of this work is obvious for robotic manipulation
- The experiments are extensive, including multiple public benchmarks. The comparison baselines are also up to date.
- The performance is strong considering that it didn’t train on the testing objects.
- To generalize to the testing object with a new coordinate frame definition provided in the CAD model, a novel and smart adaptation mechanism is designed by rendering multiple different vides to disambiguate.

Weakness:
- Is the performance equal when applied to different types of objects or is there any failure modes on certain types of objects? An illustration on this or discussion would be helpful. It’d be also interesting to have some detailed result decomposition on the objects to reflect their performance variations.
- The object detection is assumed to be available. While this is not the focus of this work, I’d imagine the accuracy of the 2D detection can play a crucial role. In addition, the current manuscript uses Mask-RCNN which needs training and inference on the same categories of objects. This contradicts with the claim of applying the method to out-of-class objects. Moreover, the Mask-RCNN is trained on the PBR data in this work, meaning many of the training instances are exactly the same as the testing data in BOP. This gives a huge advantage over the comparison methods (usually use natural detectors) from the beginning. It’d be more interesting to see a complete pipeline, by replacing with a class-agnostic object detector or at least a Mask-RCNN that is trained on some general dataset (eg. MS COO), or comparing with baselines that uses the same object detector to showcase the performance and compare.
- While the current manuscript claims to estimate 6D pose of novel objects, a CAD model is still needed during testing. The ease of accessibility to such a high-quality CAD model during testing (especially with high-fidelity texture) is questionable in real world scenarios.


**Summary Of Recommendation:**

weak accept

---

> ### Author Response · Authors · 2022-08-25
> **Thank you for your review, we respond to individual comments.**
>
> [1/2]
>
> We thank the reviewer for their feedback and provide answers to their questions below. Please note our response is split into two comments due to the character limit for a single comment.
>
> ## Failure modes and performance on specific types of objects
> The reviewer asked for some qualitative examples of failure modes and whether they are specific to certain types of objects. In Section 2 of the attached PDF, we present a per-object quantitative and qualitative analysis of our approach on the YCB-V dataset. In summary, we notice three main failure modes related to the types of objects: (i) for textureless objects with a similar appearance under viewpoints (e.g. a bowl); (ii) for asymmetric objects which “almost” look symmetric and for which it is necessary to look at very fine details to disambiguate between multiple poses (e.g. a pair of scissors with different left/right handles); and (iii) for objects which have a CAD model with a scale that do not match the real object. Please see the attached PDF for more details and illustrations. We observed the first two failure modes are caused by an inaccurate coarse estimation, which we mention as a limitation of our approach in lines 270-271 of our paper. The last failure mode is inherent to 6D pose estimation which requires access to a CAD model with the correct dimensions in order to estimate a correct and meaningful estimate.
>
> ## Assumption of object detection
> We agree that using Mask-RCNN models trained on the target objects for initial 2D detections makes it difficult to compare the performance of our approach with other methods such as Zephyr and OSOP, as mentioned in lines 212-213. We ran additional experiments where we use the same pose hypotheses as Zephyr and compared the performance of our pose estimation method. Results are presented in the table below. Averaged across the YCBV and LMO datasets we achieve an AR score of 59.7 compared to 55.7 for Zephyr.
>
> These results show that our method significantly outperforms Zephyr on YCB-V (+10.7) in a similar setup where the detections and initial pose hypotheses are obtained from a method that does not benefit from being trained on the target objects. Another significant advantage of our approach over Zephyr is that it can be run on RGB images, whereas Zephyr strongly relies on the depth modality, as mentioned in line 106 of our submission.
>
> We have also contacted the authors of OSOP in order to run a comparison of our pose estimation systems using the same detections, but we have not received a response at the time of writing this response.
>
> |                                             | LM-O     | YCB-V    | Mean     |
> |---------------------------------------------|----------|----------|----------|
> | (Drost,PPF+Sift)+Zephyr+ICP                 | **59.8** | 51.6     | 55.7     |
> | (Drost,PPF+Sift)+Our coarse+Our RGB refiner | 57.0     | **62.3** | **59.7** |
>
> ## CAD model requirement
> Our approach indeed requires - at test time - a CAD model of the novel object to estimate its pose, as explicitly stated multiple times in the paper, e.g. lines 3-4, lines 115-116. We believe there are many practical scenarios where a CAD model of an object is available: e.g. provided by the manufacturer in industrial setting, or reconstructed using a set of RGB images with Apple’s Object Capture or NeRFs, or reconstructed using a 3D scanner like the one used to reconstruct GoogleScanned objects [6]. Please also see the response to the meta-reviewer.
>
> We would also like to mention that the CAD model does not need to be high-fidelity. In Figure 4 of the attached PDF, we show examples of correctly estimated poses using low-fidelity CAD models with low-quality textures or geometric discrepancies between the real object and its 3D model.
>
> ## Clarification of Figure 2
> The blue axes illustrate the initial pose estimate $T_{CO,1}$ which is also the pose used to render the first image that is provided as input to the refinement network. The three sets of green axes illustrate the three additional viewpoints {T_{CO,i}}\_i=2^4 provided to the refinement network. These three additional viewpoints have a $z$-axis pointing towards the anchor point, while $T_{CO,1}$ doesn’t necessarily satisfy this condition. These additional views provide the refinement network information about the location of the anchor point.

---

> > ### Author Response · Authors · 2022-08-25
> > **Thank you for your review, we respond to individual comments.**
> >
> > [2/2]
> >
> > ## Pose hypothesis generation
> > The method for generating test hypotheses is described in Section 4 of the supplementary PDF of our initial submission. The $x$ and $y$ coordinates of the object translation are computed to ensure that the reprojection of the anchor point matches the center of the 2D bounding box. The depth is computed using the projection of the points of the object 3D model using a sampled random orientation and the position of the anchor point. Our method can be run with any choice of anchor point and thus does not depend on the arbitrary origin defined in the CAD model. If the origin is not within the 3D bounding box of the points of the model, an anchor point can be picked using a simple strategy like picking the centroid of the 3D points. In the 3D models we used during testing, we observed the origin typically lies within the object geometry and used this origin as the anchor point.

---

> > > ### Comment · Reviewer_KfxR · 2022-08-28
> > > **Response**
> > >
> > > Thanks for the response, it addresses my concerns

---

### Official Review · Reviewer_VhYw · 2022-08-01

**Originality:** Very Good
**Technical Quality:** Good
**Clarity Of Presentation:** Very Good
**Impact:** 3

**Recommendation:**

Weak Accept: I recommend accepting the paper, but will not argue for my recommendation if the majority of other reviewers have a different opinion.

**Summary:**

This paper study the problem of 6D pose estimation. It adopts a coarse-refine approach, where it first predicts a coarse pose hypothesis and use a refiner based render&compare to refine the pose. It can be applied to objects not seen in the training phase.

**Issues:**

1. More results to demonstrate the advantage of the coarse model over other baselines.
2. Discuss how robust the model is to sim2real gap in terms of rendering.

**Quality Of The Limitations Section:**

Limitations are addressed clearly

**Reviewer Expertise:**

2: The reviewer is willing to defend the evaluation, but it is quite likely that the reviewer did not understand central parts of the paper

**Robotics Focus:**

Sufficient demonstration on hardware

**Strengths And Weaknesses:**

Strengths:
1. The idea of using classifier to acquire coarse pose is simple and robust and it can handle symmetry implicitly.
2. The experiments demonstrate the efficacy of the refiner module.

Weaknesses:
1. The efficacy of the coarse pose estimation model is not validated. In Table 1, the comparison between 8,9.10 and 11,12,13 is unfair for the proposed coarse module. It would be stronger if there is a baseline with other pose initialization techniques that can handle novel objects and use the proposed refiner.
2. Since both the coarse mode and refiner is based on rendering, how would rendering quality and the different lighting in the real world affect the performance?


**Summary Of Recommendation:**

The general idea itself is pretty straightforward and not very interesting. But the performance of the proposed method is strong, especially on the real-world experiment. So I would vote for weak accept.

---

> ### Author Response · Authors · 2022-08-25
> **Thank you for your review, we respond to individual comments.**
>
> We thank the reviewer for their feedback and provide answers to their questions below.
>
> ## Efficacy of coarse pose estimation model
> We would like to clarify that the results in lines 8,9,10 and 11,12,13 of Table 1 aim to show that (i) our refinement network significantly improves initial coarse pose estimates regardless of the method used to obtain these initial estimates, and (ii) the performance of our refiner is competitive with the learning-based refiner of CosyPose [3] (citation numbering following the original submission) while not requiring to be trained on the test objects, as discussed in lines 217-220. Please also note that lines 12 and 13 of the table present results for our refiner applied to a pose initialization technique that can handle novel objects, which is our coarse scoring  approach in this case.
>
> In order to evaluate the performance of our coarse scoring network independently, we use a set of pose hypotheses generated for novel objects by the commercial Halcon 20.05 Progress software which implements the PPF algorithm described in [22]. Note that these are the same pose hypotheses used in Zephyr [20]. We then find the best hypotheses using the scores of PPF, the scoring network of Zephyr or our coarse network, and report AR results for the LM-O and YCB-V datasets in the table below.
>
> These results show that on both datasets, our coarse network is better than the two baselines (PPF and Zephyr) for selecting the best poses among a given set of hypotheses.
>
> |                | LM-O     | YCB-V    |
> |----------------|----------|----------|
> | PPF            | 52.7     | 34.4     |
> | PPF+Zephyr     | 59.8     | 45.8     |
> | PPF+Our coarse | **61.6** | **52.1** |
>
> ## Sim2Real gap(s)
> We would like to clarify that there are two different sim2real gaps:
>
> 1. In the observed images, between (i) the synthetic images used for training the approach which are rendered using a photorealistic renderer (e.g. Fig 1 (a) in the submission) and (ii) the real images which the approach is tested on (displayed in Fig 3 in the submission.).
> 2. For a given observed image, between (i) the observed image and (ii) the synthetic rendering of the object (e.g. shown in Fig 2.).
>
> The first sim2real gap is addressed by using photorealistic synthetic images that display realistic lightning conditions and reflections on the objects, and by using data augmentation to gain robustness with respect to the quality of the image or the illumination conditions. Using photorealistic images and data augmentation is crucial, as already shown by [1] (importance of using photorealistic images) and [3] (importance of using data augmentation). The real TUD-L dataset, which we evaluate our approach on, presents significant variations in illumination conditions. In the Section 1 of the attached pdf, we present qualitative examples of predictions on this dataset which illustrate the robustness of our approach to challenging illumination conditions in the real world.
>
> We now discuss the second sim2real gap. The same renderer is used to render the CAD model of the objects at training and test time and thus there is no “gap” that must be addressed between the training and testing phases. However, for a given novel object, the quality of the rendering of the CAD model may be limited by the quality of the CAD model itself. We do find that our method is robust to CAD models with low-quality textures and geometric discrepancies with the actual object, see Figure 4 in the attached pdf for qualitative examples. Please also see the response to the meta-reviewer and reviewer KfxR.

---

> > ### Comment · Reviewer_VhYw · 2022-08-26
> > **Response to authors' response**
> >
> > Thank you, it addressed my issues.

---

### Meta-Review · Area_Chair_sCXz · 2022-08-14

**Recommendation:** Accept (Poster)
**Confidence:** 4

**Metareview:**

Main strengths:
1. The problem setting investigated (target CAD model given during test time only) seems relevant to practical robotics applications.
2. The proposed method (binary classfier to select best coarse pose as initialisation, render-and-compare to refine pose) appears to be sensible.
3. Results are comprehensive and overall convincing enough.

Main weaknesses:
1. The premise of the setting/method (bounding box and detailed CAD model of the target object are available during test time) is not beyond questioning.
2. Comparisons against other methods are not thorough; a few useful ablation tests are missing.

After the rebuttal and discussions, all reviewers thought that the concerns have been satisfactorily addressed. The paper has thus met the threshold to be accepted.

---

> ### Author Response · Authors · 2022-08-25
> **Thank you for your review, we respond to individual comments.**
>
> We thank the meta-reviewer for their feedback. Please see the answers below.
>
> ## Requirement for object detections
> We would like to emphasize that our contribution is on a 6D pose estimation system (coarse + refiner model) that can be applied to novel objects (see lines 48-49) given a detection (2D bounding box), rather than on zero-shot detection. In the future our approach could be combined with any zero-shot object detection approach, such as those cited in lines 47 and 122. We also believe the large-scale dataset we introduce in our paper could stimulate the development of new zero-shot detection approaches.
>
> Finally, we want to mention that the 2D detections are only used in our approach to generate the initial pose hypotheses, which are then scored by our coarse scoring network. Our method can thus be used with other methods for pose hypothesis generation to form a complete pipeline. We performed new experiments to demonstrate such a complete pipeline using the pose hypotheses generated with PPF [22] on the LMO and YCBV dataset (see below, "Comparisons against prior work").
>
> ## Requirement of a CAD model
> We agree that our method requires a CAD model. However, we believe that this assumption is often satisfied in industrial settings, e.g. because it is provided by the manufacturer as discussed in line 27. For other applications where the CAD model is not available, it can be reconstructed. Several techniques are being developed to ease the process of creating textured 3D models of an object, such as (i) Apple’s Object capture, or (ii) the 3D scanner used in [6] to reconstruct the meshes of the GoogleScannedObject dataset. We thus believe that there are many realistic settings where a CAD model is available.
>
> Additionally, emerging techniques in neural graphics such as NeRFs - which provide an implicit representation of an object that can be used to render it from any viewpoint - could in the future be used instead of an explicit textured mesh model in our pose estimation pipeline based on the render-and-compare strategy.
>
> ## Comparisons against prior work
> Please note that it is difficult to directly compare our approach with OSOP and Zephyr because different strategies are used for detection or pose hypothesis generation. We discuss this point in lines 210-215 in the submission and our experiments presented in Sec 4.2 validate our contributions which are focused on the coarse and refinement strategies, and not on detection of novel objects.
>
> In order to provide a fair comparison with Zephyr, we performed additional experiments where the same pose hypotheses are used. The results presented in the responses to reviewers sFyM and KfxR show our method outperforms Zephyr on the YCB-V dataset by a significant margin when the same initial pose hypotheses are used.
> We have also contacted the authors of OSOP to get access to their 2D detections and perform additional experiments, but we did not get a reply at the time of submitting this response.